# LANGUAGE MODELING WITH GRAPH TEMPORAL CONVOLUTIONAL NETWORKS

## ABSTRACT

Recently, there have been some attempts to use non-recurrent neural models for language modeling. However, a noticeable performance gap still remains. We propose a non-recurrent neural language model, dubbed graph temporal convolutional network (GTCN), that relies on graph neural network blocks and convolution operations. While the standard recurrent neural network language models encode sentences sequentially without modeling higher-level structural information, our model regards sentences as graphs and processes input words within a message propagation framework, aiming to learn better syntactic information by inferring skip-word connections. Specifically, the graph network blocks operate in parallel and learn the underlying graph structures in sentences without any additional annotation pertaining to structure knowledge. Experiments demonstrate that the model without recurrence can achieve comparable perplexity results in language modeling tasks and successfully learn syntactic information.

## 1 INTRODUCTION

Recurrent neural networks (RNNs) are currently considered the de facto tool for language modeling tasks. RNNs encode sentences by processing each word in a sequential manner. There is a strong inductive bias, though usually implicit, that the structure of a sentence is a sequence of words. However, natural language has complex underlying syntactic structures. Although recent work (Kuncoro et al., 2018) has demonstrated that RNNs are capable of capturing structural dependencies as long as they have enough capacity, in practice, it is less clear whether RNNs are able to capture such structural information with limited resources, especially over long distances.

The underlying structures of sentences are useful in natural language understanding and generation. To fully understand the meaning of a sentence, a linguistic analysis not only needs to recognize the semantics of words in the sentence, but also needs to understand how the words are organized in the sentence. To compose a correct sentence, one should also follow grammars and syntactic rules. Unfortunately, conventional recurrent architectures are not specifically designed for modeling such structures. As a result, a poor performance can be expected for tasks requiring structural information.

Traditionally, natural language parsers are used to model grammar and syntactic knowledge. By generating a parse tree, the parser can describe the structure and word dependencies of a sentence. However, to generate such a structure often involves some form of supervised training of a parser. Recent works have demonstrated several ways of learning structural information of natural sentences without supervision, which can be helpful when performing language modeling tasks (Cheng et al., 2016; Shen et al., 2017).

In contrast to RNNs, convolutional neural network (CNNs) might be good candidates for learning structures due to their use of hierarchical filters. Recently, increasing attention has been paid to the use of CNNs for language modeling tasks. Dauphin et al. (2016) propose a gated convolutional language model, and

the experiments in Bai et al. (2018) show that temporal convolutional networks (TCNs) perform almost as well as RNNs for many tasks. However, there is a noticeable performance gap between TCNs and RNNs. Furthermore, those models are not easily interpretable in that they do not explicitly learn the structures of sentences.

A more dedicated architecture for modeling non-sequential structures is based on graph networks (GNs) (Scarselli et al., 2009). The underlying structure among nodes is described with adjacency matrices on which the model propagates messages between different nodes (Kipf & Welling, 2016; Li et al., 2016).

In this work we aim to demonstrate how a non-recurrent neural model can more explicitly model structural information in sentences without supervision for improving language modeling tasks. Specifically, this non-recurrent neural model consists of a novel convolutional architecture combined with graph network blocks. To avoid the sequential inductive bias of natural sentences, we regard the underlying structures of the sentences as fully connected graphs to learn as much syntactic information as possible. We also apply a message propagation mechanism to learn the representations of words and sentences. Though individual modules used in this paper have been previously reported, to our knowledge, the combination of these components to achieve the reported results on language modeling tasks is novel.

The contributions of this work are as follows:

1. We show how RNNs model implicit connections between words, though they carry a sequential inductive bias.

2. We propose a graph- and convolution- based language model, which can run in parallel. To our knowledge, the model achieves state-of-the-art results on the Penn Treebank dataset compared to existing non-recurrent language models.

3. We introduce an unsupervised approach to learn syntactic parsing structure and other dependencies among words, which further improves the performance of our proposed language model.

## 2 RELATED WORK

RNNs have been widely applied to language modeling tasks for over a decade. For example, Cheng et al. (2016) use a long short-term memory (LSTM) RNN to learn a self-attention mechanism that indicates contextual dependencies in a language modeling task. As another example, Shen et al. (2017) propose an unsupervised learning method for syntactic distance, in which the language model can generate a parsing tree of a sentence. To achieve faster training speed, Gao et al. (2018) adopts a group strategy for recurrent layers. Also, inspired by human speed reading mechanisms, several RNN variants (Seo et al., 2017; Campos et al., 2017) have been proposed to decide how to update their hidden states based on the importance of input tokens in a sequence. In contrast to RNN-based language models, it has been recently argued that temporal convolutional networks (TCNs) have competitive performance in some NLP tasks including word-level and character-level language modeling (Bai et al., 2018). As is the case for RNNs, TCNs cannot easily model relational information well by design.

Many kinds of sequential data can be represented as graphs, in which nodes stand for entities and edges stand for relations. In order to model relational structures, our framework is based on graph neural networks (GNNs), which were first proposed in (Scarselli et al., 2009). Subsequently, the authors in Li et al. (2016) replaced the Almeida-Pineda algorithm used in (Scarselli et al., 2009) with backpropagation. In (Garcia & Bruna, 2018) GNNs were applied to image classification tasks in a "few-shot" setting. In (Gilmer et al., 2017), GNNs were applied to molecular property prediction tasks. The authors also summarized and generalized a message-passing mechanism. Kipf & Welling (2016) proposed graph convolution networks (GCNs) for semi-supervised learning, which conduct message propagation without learning edge representations. Li et al. (2016) introduce a gating mechanism, which improves the performance of GNNs. Unfortunately,

there are relatively few papers discussing how to adapt GNNs to natural language tasks. Marcheggiani & Titov (2017) applied GNNs to semantic role labeling. Schlichtkrull et al. (2017) used GNNs to perform knowledge base completion tasks. (Johnson, 2017) generated a graph based on textual input and update the relationship during training. Scarselli et al. (2009) proposed a neural model for message propagation in graphs, using an adjacency matrix, $\mathbf{A}$, of a target graph to maintain structural information. Finally, (Seo et al., 2016) combined GNNs with RNNs for graph structure data.

## 3 BACKGROUND

### 3.1 LONG SHORT-TERM MEMORY AND VARIANTS

LSTM (Hochreiter & Schmidhuber, 1997) attempts to solve gradient problems associated with training recurrent networks and memorize longer sequences. The basic feedforward propagation method of LSTMs is shown as follows:

$$
\begin{aligned}
\mathbf{z}_t &= tanh(\mathbf{W}_z^x \mathbf{x}_t + \mathbf{W}_z^h \mathbf{h}_{t-1} + \mathbf{b}^z) \\
\mathbf{f}_t &= \sigma(\mathbf{W}_f^x \mathbf{x}_t + \mathbf{W}_f^h \mathbf{h}_{t-1} + \mathbf{b}^f) \\
\mathbf{o}_t &= \sigma(\mathbf{W}_o^x \mathbf{x}_t + \mathbf{W}_o^h \mathbf{h}_{t-1} + \mathbf{b}^o) \\
\mathbf{i}_t &= \sigma(\mathbf{W}_i^x \mathbf{x}_t + \mathbf{W}_i^h \mathbf{h}_{t-1} + \mathbf{b}^i) \\
\mathbf{c}_t &= \mathbf{f}_t \odot \mathbf{c}_{t-1} + \mathbf{i}_t \odot \mathbf{z}_t \\
\mathbf{h}_t &= \mathbf{o}_t \odot tanh(\mathbf{c}_t)
\end{aligned}
\tag{1}
$$

A variant of LSTMs, named Quasi-RNNs (QRNN) (Bradbury et al., 2016), calculates $\mathbf{z}, \mathbf{f}, \mathbf{o}, \mathbf{i}$ in parallel by replacing $\mathbf{h}_{t-1}$ with $\mathbf{x}_{t-1}$, which significantly accelerates training. However, QRNNs still have to calculate cell values sequentially.

Applying the inductive bias that the underlying structure of natural sentences are linear sequences, LSTMs only explicitly model sequential structures. The relation between two words with a distance larger than 1 is modeled implicitly with a sequence of input and forget gates. Given equation 1, we can rewrite the message propagation in LSTM cells in the form of $\mathbf{c} = \mathbf{A} \cdot \mathbf{z}$, where $\mathbf{c} = [c_1, c_2, \ldots, c_t]^T$, and $\mathbf{z} = [z_1, z_2, \ldots, z_t]^T$ as follow:

$$
\mathbf{c} = \underbrace{\begin{bmatrix} i_1 & & & \\ f_2 i_1 & i_2 & & \\ \cdots & \cdots & & \\ \prod_{i=2}^t f_i \cdot i_1 & \prod_{i=3}^t f_i \cdot i_2 & \cdots & i_t \end{bmatrix}}_{\mathbf{A}} \begin{bmatrix} z_1 \\ z_2 \\ \cdots \\ z_t \end{bmatrix}
\tag{2}
$$

In the above equation, each element in $\mathbf{A}$, $\mathbf{A}_{ij}$ indicates the amount of message passed from $z_j$ to $z_i$, where

$$
a_{ij, i \neq j} = \prod_j^i f_{j+1} \cdot i_j
\tag{3}
$$

$$
a_{ii} = i_i
\tag{4}
$$

Inspired by recent work on graphical neural models (Scarselli et al., 2009; Kipf & Welling, 2016; Veličković et al., 2017), we can regard $\mathbf{A}$ as the adjacency matrix of a weighted, directed graph $\mathbf{G} = (\mathbf{z}, \mathbf{E})$, where $\mathbf{z} = [z_1, z_2, \ldots, z_t]^T$ are nodes and each edge $\mathbf{e} \in \mathbf{E}$ indicates the amount of information needed to be propagated from one node to another. Equation 2 shows how LSTMs implicitly learn relation between each pair of words in a sentence.

## 3.2 GRAPH NEURAL NETWORKS

We argue that sentences can be structured as graphs, in which nodes stand for entities and edges stand for relations. Following (Scarselli et al., 2009; Kipf & Welling, 2016), the message propagation in a GCN is given by:

$$\mathbf{X}^{l+1} = f(\mathbf{D}^{-\frac{1}{2}}\mathbf{A}\mathbf{D}^{-\frac{1}{2}} \cdot \mathbf{X}^l \mathbf{W}) \tag{5}$$

where $\mathbf{x}^l$ and $\mathbf{x}^{l+1}$ are the input and output of the current layer, $f$ is a non-linear activation, $\mathbf{D}$ is the diagonal degree matrix of adjacency matrix $\mathbf{A}$, and $\mathbf{W}$ stands for a learnable weight matrix in the current layer.

The gating mechanism can also be applied to message propagation in graphical models. Li et al. (2016) propose gated graph neural networks (GGNNs), which filter the embeddings of the target node and its neighbors with gates instead of simply averaging them as in GCNs.

## 4 GRAPH TEMPORAL CONVOLUTIONAL NETWORKS

Inspired by the analysis of LSTMs and recent work on graphical neural models, in this section we introduce graph temporal convolutional networks (GTCNs) for language modeling. We are motivated by the fact that we can calculate the adjacency matrix, $A$, in parallel, and thus avoid the difficulties of training recurrent neural structures and adopting the sequential hypothesis for language models. The GTCN model consists of two modules: context attention and message propagation.

### 4.1 POSITION-AWARE CONTEXT ATTENTION

To decide how much the previous words contribute to predicting the next word, the GTCN model generates an attention over a context for each target word. Given the input sequence consists of $n$ words, $\mathbf{X} = [\mathbf{x}_1, \mathbf{x}_2, \ldots, \mathbf{x}_n]$, we calculate key and value vectors, $\mathbf{k}$ and $\mathbf{v}$, for each word for calculating the attention,

$$\mathbf{q}_i = tanh(\mathbf{W}_1^q \cdot \mathbf{x}_i + \mathbf{W}_2^q \cdot \mathbf{x}_{i-1} + \mathbf{b}^q) \tag{6}$$

$$\mathbf{k}_i = tanh(\mathbf{W}_1^k \cdot \mathbf{x}_i + \mathbf{W}_2^k \cdot \mathbf{x}_{i-1} + \mathbf{b}^k) \tag{7}$$

where $*$ is a convolution operation.

For words $\mathbf{x}_i$ and $\mathbf{x}_j (j < i)$, we calculate their relation $a_{ij}$ as follows,

$$a_{ij} = \frac{e_{ij}}{\sum_{s<k<i} e_{ik}} \tag{8}$$

$$e_{ij} = \exp(\mathbf{k}_i \cdot (\mathbf{v}_j + \mathbf{W}_{i-j}^p) + \mathbf{b}^e) \tag{9}$$

where $s$ is the start location of a visible window, and $\mathbf{W}_d^p$ is the relative position representation proposed in (Shaw et al., 2018), which encodes the distances between different words.

### 4.2 MESSAGE PROPAGATION

We apply the gated graph neural network (GGNN) architecture described in Li et al. (2016) for message propagation in each layer of the GTCN. In contrast to the original GGNN, the GTCN message propagation method does not have a recurrent architecture. Given input sequence $\mathbf{X} = [\mathbf{x}_1, \mathbf{x}_2, \ldots, \mathbf{x}_t]^T$, we summarize the entire sentence with generated attention to predict $\mathbf{x}_{t+1}$. The context representation $\mathbf{x}_t^c$ is calculated as follows:

$$\mathbf{c}_t = \sum_i^{t-1} a_{ti}\mathbf{x}_i \tag{10}$$

We then calculate input gate $\mathbf{i}$, forget gate $\mathbf{f}$, and residual gate $\mathbf{r}$ with $\mathbf{x}_t$ and $\mathbf{x}_t^c$,

$$\mathbf{g} = \sigma(\mathbf{W}_1^g\mathbf{x}_t + \mathbf{W}_2^g\mathbf{c}_t + \mathbf{b}^g), \mathbf{g} \in [\mathbf{i}, \mathbf{f}, \mathbf{r}] \tag{11}$$

We also calculate an output gate for the GTCN,

$$\mathbf{o} = \sigma(\mathbf{W}_1^o\mathbf{x}_t + \mathbf{W}_2^o\mathbf{x}_{t-1} + \mathbf{b}^o) \tag{12}$$

The GTCN predicts the embedding of $\mathbf{x}_{t+1}$ as

$$\mathbf{m}_t = \mathbf{o} \odot \phi(\mathbf{W}_1^h \cdot (\mathbf{x}_t \odot \mathbf{i}) + \mathbf{W}_2^h \cdot (\mathbf{c}_t \odot \mathbf{f}) + \mathbf{b}^h) \tag{13}$$

$$\hat{\mathbf{x}}_{t+1} = \mathbf{h}_t = \mathbf{i} \odot \mathbf{m}_t + (\mathbf{1} - \mathbf{i}) \odot \mathbf{m}_t \tag{14}$$

where $\odot$ stands for element-wise product, and $\phi$ is a non-linear activation function.

### 4.3 SYNTAX LEARNING AND REPRESENTATION

Given a sentence $X = [x_1, x_2, \ldots, x_n]$, the GTCN model generates a attention weights for every pair of words. In other words, for each word $x_t$, there is a corresponding attention weight sequence $[a_1, a_2, \ldots, a_{t-1}]$. The attention weights are calculated with equation 8 described above. This feature makes our approach different from the syntactic distance model (Shen et al., 2017), which learns one parsing tree for an entire sentence.

Similar with recurrent neural network grammars (RNNG) (Dyer et al., 2016), our model constructs a unique parsing tree for each word in input sentences with other visible tokens. Experiment shows that the attention weights generated by GTCN captures information about the structure of the ground truth parse tree.

### 4.4 OTHER DETAILS

**Convolution Windows** We hope the multi-layer GTCN model is able to learn different levels of semantics in the given sentence. In our design of the model, the lower GTCN layers learn local information, while the upper layers learn information from a longer context. In practice, the size of the context window of layer $i$ is $i \cdot L$, where $L$ is the context window length of the first GTCN layer. Experiments show that this strategy works better than making the entire sentence visible for all layers and using the same context window size for each layer.

| Model | PPL | Model Size | Recurrence | Syntax |
|---|---|---|---|---|
| LSTM (Zaremba et al., 2014) | 78.4 | 66M | Yes | - |
| Variational LSTM (Gal & Ghahramani, 2016) | 73.4 | 66M | Yes | - |
| LSTM + continuous cache pointer (Grave et al., 2016) | 72.1 | - | Yes | - |
| Variational RHN (tied) (Zilly et al., 2016) | 65.4 | 23M | Yes | - |
| 11-layer IndRNN (Li et al., 2018) | 65.3 | - | Yes | - |
| AWD-3-layer LSTM (tied) (Merity et al., 2017) | **57.3** | 24M | Yes | - |
| TCN (Bai et al., 2018) | 88.7 | - | - | - |
| CharCNN (Kim et al., 2016) | 78.9 | 19M | - | - |
| **GTCN** (ours) | **66.1** | 27M | - | Yes |
| RNNG (Dyer et al., 2016) | 102.4 | - | Yes | Yes |
| PRPN (Shen et al., 2017) | **62.0** | - | Yes | Yes |

Table 1: Evaluation of different models on word-level language modeling in terms of test perplexity (PPL).

**Variational Dropout** The standard dropout model (Srivastava et al., 2014) randomly samples masks for input tensors at each time step in sequence processing. In our GTCN model, we employ variational dropout (Gal & Ghahramani, 2016), which samples one mask for the same variables in different time steps, to improve training convergence.

## 5 EXPERIMENTS

The experiments are designed to answer the following questions: **Q1:** How effective is the graph convolutional language model, compared with RNN-based language models? **Q2:** Can the GTCN language model learn syntactic information without any annotations?

### 5.1 SETTINGS

We conduct experiments on language modeling (LM) on the Penn Treebank (PTB) dataset (Mikolov et al., 2010), which includes 10,000 different words. Our GTCN model employed 4 layers, in which each layer applies a convolution window sized 10, 20, 30, and 40 respectively. The embedding size of words is chosen as 400, and the hidden layers were 800 dimensions. We also applied tied weights for the encoder and decoder.

In the optimization process, we apply stochastic gradient descent (SGD) and the average-SGD (ASGD) strategy proposed in (Merity et al., 2017). The batch size is 20, and the length of training sequences is 70. The initial learning rate of the SGD step is 30. The dropout rate of the embedding layer is $0.4$, while the hidden layers apply $0.25$.

### 5.2 EFFECTIVENESS (Q1)

Table 1 shows the performance of our model and lists a selection of recent baselines. The experiments show that our model outperforms many strong baselines, but is not as good as the current best LSTM language models. However, our model outperforms other CNN-based model significantly, decreasing the perplexity of the TCN model by over 20 points. To our knowledge, this is the first convolution based neural language model that has reached such performance on the PTB dataset. We also compare our model with prior work that make use of syntactic information in language modeling. The GTCN model performs better than all

models in this family except the parse-read-predict network (PRPN) (Shen et al., 2017), which employed an LSTM architecture with semantic and syntactic attention.

### 5.3 LEARNING STRUCTURAL INFORMATION (Q2)

In this section, we examine the structural information learned by the GTCN language model with an example. By taking the example "I shot an elephant in my pajamas[1]" as the input of a trained 4-layer GTCN language model, we can visualize the attentions generated by GTCN while encoding the sentence, for predicting the words, on each layer of the network.

Figure 1 shows two parsing trees of the example sentence. In this example, the parsing tree on the left makes more sense because "I" am more likely to be "in my pajamas". Admitted, both are possible. With this example, we show how the attention mechanism works in GTCN.

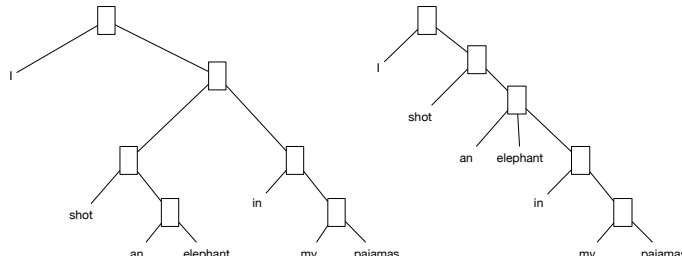

Figure 1: Two possible parse trees for the sentence "I shot an elephant in my pajamas."

To analyze the syntactic information learned by the model, we visualize the attentions generated while processing each word. For each word, we plot its attention weights on all GTCN layers.

Figure 2 visualizes attentions generated when the GTCN model processes the input sentence. The target word processed at each step is shown in red. Slashed edges stand for words and nodes that are not visible at the current step. The arrows are the edges of temporal parse trees constructed only with visible words. We color the visible nodes in the parse trees according to attentions shown above.

Since we calculate attention weights for every pair of words, the GTCN can use different syntactic information while processing the sentence at different steps for better encoding target words. For example, when processing the word "elephant", the model highlights "shot" and "an". In the fourth parsing structure, "elephant" and "in" receive higher weights when the model encodes word "my". In the third tree, while processing word "in", the model assigns higher weights on "shot", and "elephant", but pays less attention to "shot". This suggests that the model decides that "in" is leading a word sequence that describes "an elephant". This prediction indicates that the model prefers the second parse tree when encoding the sentence.

Although the model generates different attentions on context while processing different words, we find a common phenomenon of the learned attention weights. As illustrated in Figure 2, the attention mechanism learns a path from target words to the root (PTR) of the parse trees. In general, the nodes on this path are assigned higher attention weights. While modeling the word "in", the model decides that "an elephant" contains enough information to encode "in", so it does not use much higher level information.

The above example illustrates that our model is more flexible than CNNs and LSTMs in learning structures. CNNs have a fixed kernel size and dilation size (Bai et al., 2018). The structure of context modeling in CNNs totally depends on hyper-parameter settings, including kernel size, dilation size, and the number of

---

[1] https://www.nltk.org/book/ch08.html

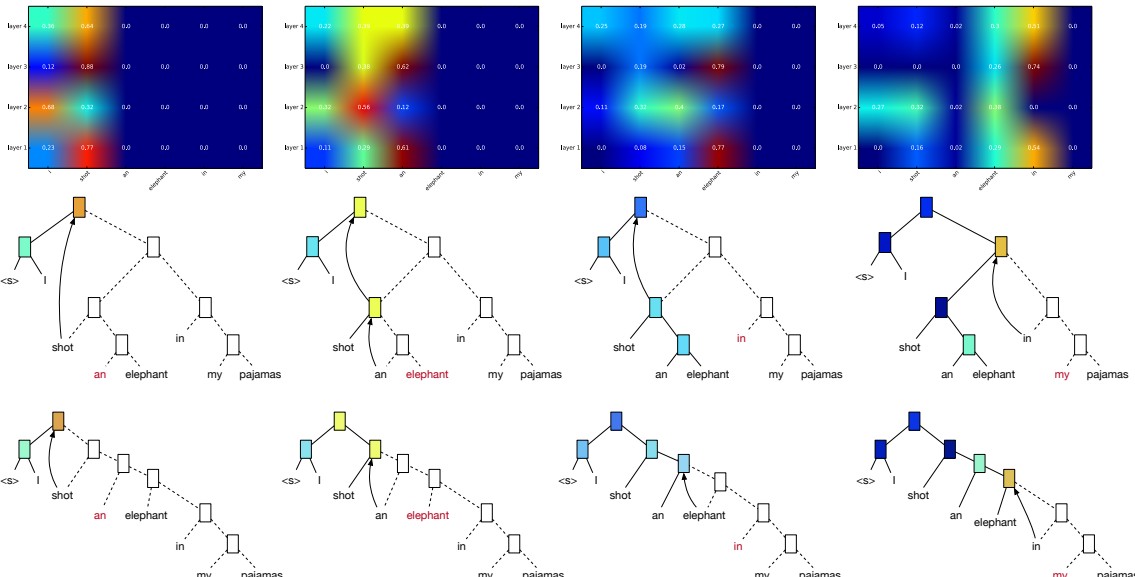

Figure 2: Attentions generated when processing target words marked with red. The upper plots show the attention weights, and the lower figures show the possible underlying syntactic structures.

layers. In contrast, LSTMs process sentence sequentially, and the relation between words $i$ and $j$, $a_{i,j}$, is always smaller than $a_{i+1,j}$. With our model, which does not employ fixed-length filters or recurrent cells, the language model can learn more information about context structures, such as PTR, for better predictions.

## 6 CONCLUSION AND FUTURE WORK

In this work, we proposed a graph temporal convolution network (GTCN) for language modeling that processes natural sentences as graphs instead of linear sequences. Without any recurrent components, our proposed model can run in parallel. Although there is still a performance gap between our model and state-of-the-art RNN-based LMs, our GTCN LM significantly outperforms existing convolution-based models. We also proposed the right-root parsing trees (RPTs) for representing syntactic information of sub-sentences in language modeling tasks. By visualizing the attention weights generated by the GTCN and constructing the RPTs, we illustrated that the GTCN can capture syntactic information for language modeling. We believe that due to the ability to run in parallel and capturing syntactic information, it is worth exploring further how the GTCN model performs on other natural language processing applications that need sentence encoding with syntactic knowledge.

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
