# OpenReview forum: "Language Modeling with Graph Temporal Convolutional Networks"
_ICLR.cc/2019/Conference_

### Official Review · AnonReviewer2 · 2018-11-01

**Rating:** 4
**Confidence:** 5

**Review:**

The paper applies graph convolutional networks to Penn Treebank language modeling and provides analysis on the attention weight patterns it uses.

Clarity: the paper is very clearly written!

The introduction states that existing CNN language models are "not easily interpretable in that they do not explicitly learn the structures of sentences". Why is this? The model in this paper computes attention values which is interpreted by the authors as corresponding to the structure of the sentence but there are equivalent means to trace back feature computation in other network topologies as well.

My biggest criticism is that the evaluation is done on a very small language modeling benchmark which is clearly out of date. Penn Treebank is the CIFAR10 of language modeling and any claims on this dataset about language modeling are highly doubtful. Models today have tens and hundreds of millions of parameters and training them on 1M words is simply a regularization exercise that does not enable a meaningful comparison of architectures.

The claims in the paper could be significantly strengthened by reporting results on at least a mid-size dataset such as WikiText-103, or better even, the One Billion Word benchmark.

---

> ### Author Response · Authors · 2018-11-25
> **CNNs and datasets**
>
> Thanks for the insightful comments. CNNs process sentences by gathering larger and larger contexts in each layer, but do not explicitly model the relations among different words. That is why we said they are not easily interpretable and do not explicitly learn the structures of sentences. Our model explicitly learns the relations among words and we can interpret the relations with attention weights.
>
> About datasets - we did not try to claim that our model is the state-of-the-art LM. Testing the model on the PTB dataset, we are trying to make the points,
>
> 1. The GTCN model, which is not recurrent, is as good as many recurrent neural models
> 2. The relations among words can be explicitly modeled without supervision
>
> We believe your suggestions on using larger datasets is valuable. We will test our model on larger corpora in our future works.

---

### Official Review · AnonReviewer3 · 2018-11-01
**Interesting but requires more experiments**

**Rating:** 4
**Confidence:** 3

**Review:**

This work proposes a CNN based language model based on graph neural networks. Basic idea is to compute adjacency matrix for an entire sentence in parallel for faster computation. Empirical results show probably the best performance among CNN approaches but still lags behind the best RNNs.

Pros:

- A new network based on graph neural networks.

Cons:

- The proposed model needs to recompute attention probabilities for each step and it might incur latencies. I'd like to know how slow it is when compared with other CNN approaches and how fast it is when compared with other RNNs.

- Lacking experiments. This paper shows only a single table comparing other approaches, and does not present any ablation studies. Note that section 4.4 mentions some details, but does not show any numbers to justify the claim, e.g., why choosing the window size of 10, 20, 30, 40.

- This paper claims that the learned model captures the ground truth parse tree in section 4.3. However, this work simply picks a single example in section 5.3 to justify the claim. I'd recommend the author to run a parser to see if the proposed attention mechanism actually capture the ground truth parse trees or not.

---

> ### Author Response · Authors · 2018-11-25
> **About experiments**
>
> Thanks for the review and comments.
>
> Firstly, we did not compare the time complexity since the LSTM model in pytorch was not implemented in python, which made it difficult to compare. In the revised version, we will implement a python LSTM and compare the time it takes to train all the different models.
>
> Secondly, thank you for your suggestions on ablation studies. We chose the window size by testing the model on the dev set.
>
> About the parsing structure - our model is not able to capture the exact parsing trees. With the example in the paper, we attempted to qualitatively indicate that the attentions revealed some syntactic knowledge. We the learned attention, it still needs careful design of the algorithm to find a best parsing, which will be included in future works.

---

### Official Review · AnonReviewer1 · 2018-11-02
**A well-motivated work, but relations to prior works need to be addressed**

**Rating:** 4
**Confidence:** 5

**Review:**

This paper draws inspiration from recent works on graph convolutional networks and proposes GTCN, a convolutional architecture for language modeling. The key intuition is to treat sentences as (potentially densely-connected) graphs over tokens, instead of sequences as in many RNN-based language models. The model then, when predicting a token, summarizes previous tokens using attention mechanism as context. Empirical evaluation on word-level language modeling on Penn Treebank shows competitive performance.

The idea of this work appears reasonable and well-motivated to me. But the connections to previous works, especially those based on self-attention, should be clearly addressed. Further, writing can be improved, and I would encourage a thorough revision since there are typos making the paper a bit hard to follow.
Last but not least, I find several of the claims not very-well supported. Please see details below.

Pros:
- Well-motivated intuition treating language as structured.

Cons:
- Writing can be improved.
- Missing discussion of existing works.

Details:

- Based on my understanding of Eqs. 6--11, the proposed GTCN seems to be a gated version (also equipped with window-2 convolutions) of the self-attention mechanism. Could the authors comment on how GTCN relates to Vaswani et al. (2017), Salton et al. (2017), among others? Also, empirical comparisons to self-attention based language models might be necessary.

- I was confused by Eqs. 13--14 and the text around it. Doesn't one need some kind of classifier (e.g., an MLP) to predict x_{t+1}? Why are these two equations predicting word embedding?

- The start of Section 4.1. There seems to be a typo here. I'm assuming the two vectors are `$\mathbf{v}$ and $\mathbf{q}$` here, as in Eqs. 6 and 7.

- More clarification on Eq. 9 might be necessary. Is \mathbf{W}^p part of the parameters? I'm guessing \mathbf{W}_{i-j}^p selects a row from the matrix, since there is a dot product outside.

- Can the authors clarify Eq. 5? I'm not sure how to interpret it, and it seems not used anywhere else.

- Eq. 2 is a bit misleading: it might give the impression that f_{t+1} does not depend on f_t (and so forth), which is not the case for LSTM.

- It would be interesting to be how GTCN compare to other models in efficiency, since the paper mentions parallel computation many times.

- Contribution.2: GTCN is not really the state-of-the-art model on LM.

- Comparison to RNNG: RNNG treats each sentence as a separate sequence, in contrast to most cited works in Table 1, where the whole training (eval) set is treated as a single sequence, and truncate the length when applying BPTT. And according to the second paragraph of Section 5.1, this work follows the latter. To the best of my knowledge, such a difference does have an effect on the perplexity metric. In this sense, RNNG is not comparable to the rest in Table 1. It is perhaps fine to still put it in the table, but please clarify it in the text.

Minors:

- Why is the margins above equations seem larger. Can the authors make sure the template is right?

- Around Eq.5: why is \mathbf{X} is capitalized in the eq, but not in the text? Are they the same thing?

- Section 4.3: the dependence of attention weights $a$ is not reflected in the notation.

- Section 5.1: I think what it means here is a `10K` vocabulary, instead of a 10K word tiny corpora.


References

Vaswani et al.. 2017. Attention is All You Need. In Proc. of NIPS.

Salton et al.. 2017. Attentive Language Models.

---

> ### Author Response · Authors · 2018-11-25
> **Summarization of related previous works**
>
> Thanks for the careful review and detailed comments. In this comment I majorly explain the relations between our model and the transformer model (Vaswani et al. 2017).
>
> We are aware of the transformer model and related works in machine translation, but I did not find any previous work that directly applied transformers on language modeling on the PTB dataset. In terms of the model itself, our model applied a gated self-attention to "simulate" LSTMs. The architecture was purely motivated by LSTMs. On the other hand, we believe the most significant difference between our model and transformer is that transformers apply multi-head attentions, and the GTCN only uses one attention. The multi-head attention mechanism brings a huge number of parameters, compared with the GTCN model.
>
> About Eq. 13-14 - we used tied output embeddings (Ofir and Wolf, 2016). The output hidden state of the Eq 14 is used to be compared with the word embeddings of the decoders. That's why we said the model predicted the embedding of the next word.
>
> Eqs. 6 and 7 included typos. Thank you for reminding!
>
> In Eq. 9, \mathbf{W}^p is part of the parameters, and i-j means a line of it. I will make more explanations in the revised version.
>
> Eq. 5 stands for a normalization method proposed in a related work. We did not use this in GTCN, but it could be. In practice we use softmax attention to normalize the context information.
>
> In Eq. 2, f_{t+1}  does rely on f_t. We put the equation here to indicate in LSTM language models, how a context word influences a target word by calculating a weight between them. This motivates the proposal of the GTCN model.
>
> In terms of time complexity, we believe that processing sentences in parallel is more efficient than recurrent models. We did not compare the exact time of the GTCN and LSTM because the LSTM in pytorch is not implemented in Python. We will try to implement python LSTM and compare the time complexities of both models.
>
> GTCN is not really the state-of-the-art LM, but in our comparisons, it performs best among the non-recurrent models.
>
> About the minor questions,
>
> We had a problem in templates because we used the geometry package and messed everything. We will correct this in the revised version.
>
> In Eq. 5, X stands for a matrix that includes all nodes, while x means the embedding of one node.
> And we are using a 10K vocabulary corpus, instead of 10K words.

---

### Meta-Review · Area_Chair1 · 2018-12-16
**interesting direction but not ready for publication**

**Confidence:** 5
**Recommendation:** Reject

**Metareview:**

Though the overall direction is interesting,  the reviewers are in consensus that the work is not ready for publication (better / larger scale evaluation is needed, comparison with other non-autoregressive architectures should be provided, esp Transformer as there is a close relation between the methods).